# Phytochemical Composition and Pharmacological Activities of Three Essential Oils Collected from Eastern Morocco *(Origanum compactum*, *Salvia officinalis*, and *Syzygium aromaticum)*: A Comparative Study

**DOI:** 10.3390/plants12193376

**Published:** 2023-09-25

**Authors:** El Hassania Loukili, Safae Ouahabi, Amine Elbouzidi, Mohamed Taibi, Meryem Idrissi Yahyaoui, Abdeslam Asehraou, Abdellah Azougay, Asmaa Saleh, Omkulthom Al Kamaly, Mohammad Khalid Parvez, Bouchra El Guerrouj, Rachid Touzani, Mohammed Ramdani

**Affiliations:** 1Laboratory of Applied Chemistry and Environment, Faculty of Sciences, Mohammed First University, Oujda 60000, Morocco; ouahabi.safae@ump.ac.ma (S.O.); r.touzani@ump.ac.ma (R.T.); moharamdani2000@yahoo.fr (M.R.); 2Centre de l’Oriental des Sciences et Technologies de l’Eau et de l’Environnement (COSTEE), Mohammed First University, Oujda 60000, Morocco; elguerroujb@gmail.com; 3Laboratoire d’Amélioration des Productions Agricoles, Biotechnologie et Environnement (LAPABE), Faculty of Sciences, Mohammed First University, Oujda 60000, Morocco; amine.elbouzidi@ump.ac.ma; 4Laboratory of Bioresources, Biotechnology, Ethnopharmacology and Health, Faculty of Sciences, Mohammed First University, Oujda 60000, Morocco; iy.meryem@ump.ac.ma (M.I.Y.); asehraou@yahoo.fr (A.A.); 5Laboratory of Applied Geosciences (LGA), Faculty of Sciences, Mohammed First University, Oujda 60000, Morocco; a.azougay@ump.ac.ma; 6Department of Pharmaceutical Sciences, College of Pharmacy, Princess Nourah bint Abdulrahman University, P.O. Box 84428, Riyadh 11671, Saudi Arabia; asali@pnu.edu.sa (A.S.); omalkmali@pnu.edu.sa (O.A.K.); 7Department of Pharmacognosy, College of Pharmacy King Saud University, P.O. Box 3660, Riyadh 11481, Saudi Arabia; mohkhalid@ksu.edu.sa

**Keywords:** biological activities, essential oils, phytochemical composition, GC/MS, antioxidant activity, antimicrobial activity, molecular docking

## Abstract

Throughout history, essential oils have been employed for their pleasing scents and potential therapeutic benefits. These oils have shown promise in various areas, including aromatherapy, personal care products, natural remedies, and even as alternatives to traditional cleaning agents or pest control solutions. The study aimed to explore the chemical makeup, antioxidant, and antibacterial properties of *Origanum compactum* Benth., *Salvia officinalis* L., and *Syzygium aromaticum* (L.) Merr. et Perry. Initially, the composition of the three essential oils, *O. compactum* (HO), *S. officinalis* (HS), and *S. aromaticum* (HC) was analyzed using GC-MS technology, revealing significant differences in the identified compounds. α-thujone emerged as the predominant volatile component in the oils, making up 78.04% of the composition, followed by eugenol, which constituted 72.66% and 11.22% of the HC and HO oils, respectively. To gauge antioxidant capabilities, tests involving DPPH scavenging capacity and total antioxidant capacity were conducted. Antioxidant activity was determined through the phosphomolybdate test and the DPPH• radical scavenging activity, with the HO essential oil displaying significant scavenging capacity (IC_50_ of 0.12 ± 0.02 mg/mL), similar to ascorbic acid (IC_50_ of 0.26 ± 0.24 mg/mL). Similarly, the TAC assay for HO oil revealed an IC_50_ of 1086.81 ± 0.32 µM AAE/mg. Additionally, the oils’ effectiveness against four bacterial strains, namely *Escherichia coli*, *Pseudomonas aeruginosa*, *Staphylococcus aureus*, and *Listeria monocytogenes*, and five fungi, *Geotrichum candidum*, *Aspergillus niger*, *Saccharomyces cerevisiae*, *Candida glabrata*, and *Candida albicans*, was tested in vitro. The examined essential oils generally exhibited limited antimicrobial effects, with the exception of HC oil, which demonstrated an exceptionally impressive level of antifungal activity. In order to clarify the antioxidant, antibacterial, and antifungal effects of the identified plant compounds, we employed computational methods, specifically molecular docking. This technique involved studying the interactions between these compounds and established protein targets associated with antioxidant, antibacterial, and antifungal activities.

## 1. Introduction

Aromatic plants, also known as aromatic herbs, possess a distinct fragrance due to volatile aromatic compounds, such as essential oils, in their leaves, flowers, stems, or other plant parts [1]. These compounds are responsible for these plants’ characteristic scents and flavors [1]. Due to their bioactive characteristics and therapeutic potential, medicinal and aromatic plants have attracted significant interest in the pharmaceutical industry [2,3]. They are abundant in natural chemical compounds, including volatile organic compounds, flavonoids, terpenoids, and other chemicals that give rise to various pharmacological effects [4,5]. Aromatic plants produce essential oils, which are highly fragrant and contain the plant’s distinctive aroma. These plants are commonly used in various industries, including perfumery, cosmetics, culinary arts, and traditional medicine [6,7]. Many aromatic plants also possess medicinal properties and have been used for centuries in herbal medicine worldwide [8]. Many aromatic and medicinal plants have antibacterial properties [9], inhibiting the growth of bacteria, fungi, and viruses. These plants can potentially be employed in disease prevention and as a source of novel antibacterial chemicals [10,11]. The volatile organic compounds have been investigated for their potential antioxidant properties due to the presence of various bioactive compounds such as phenols, terpenes, and flavonoids [12]. Antioxidants help protect cells from damage caused by harmful molecules called free radicals, associated with oxidative stress and various chronic diseases [13]. Their effectiveness as antioxidant agents in the human body is still being researched. The efficacy and interest of essential oils and their importance in pharmacology depend on the extraction method, the plant source and chemical composition, as well as the presence of volatile compounds responsible for the plant’s aroma and flavor [10,14]. These pharmacological interests of aromatic and medicinal plants are the subject of extensive pharmacological research. Many studies focus on identifying bioactive compounds, mechanisms of action, toxicity, and potential drug interactions [15,16,17]. However, it is important to stress that using these plants for therapeutic purposes requires rigorous evaluation.

In this systematic paper, we discuss the chemical composition of essential oils from three medicinal plants, *Origanum compactum* (HO), *Salvia officinal* (HS), and *Syzygium aromaticum* (HC). However, no investigations on the content and biological activity of (HO), (HS), and (HC) oils, particularly those harvested in northern Morocco, are known. As a result, the study sought to determine the phytochemical content, as well as the antioxidant and antibacterial activity, of (HO), (HS), and (HC). A computational analysis was used to assess the physicochemical attributes of each detected molecule in the oils, including drug-likeness and pharmacokinetic qualities. In addition to in vitro investigations, a computational study was carried out using molecular docking and an ADMET analysis. The main purpose of the molecular docking experiment was to determine how the ligands interacted with the individual proteins involved in the investigated biological processes. This investigation provided critical information about the binding mechanisms, affinities, and potential interactions between the ligands and the target proteins. The ADMET analysis, on the other hand, attempted to evaluate the three essential oils’ various pharmacokinetic features, providing significant information about their absorption, distribution, metabolism, and other relevant aspects.

## 2. Results

### 2.1. Phytochemical Analysis

Studying the chemical composition of any plant is a highly useful way to assess its quality before its use or consumption. In this context, the chemical composition of the essential oils from three plants, namely *S. officinalis*, *O. compactum*, and *S. aromaticum*, was investigated, and the results are presented in Table 1 and Figure 1 and Figure 2. The results show that the essential oil obtained with hydrodistillation of *S. officinalis* contains eleven compounds, with the major ones being α-thujone and p-cymene, accounting for 78.04% and 17.77%, respectively.

In *O. compactum*, the major compound identified was thymol, accounting for (37.68%), followed by carvacrol (12.73%) and *p*-cymene (13.33%). As for the essential oil of *S. aromaticum*, the main identified components were eugenol, with a concentration of 72.66%, and β-caryophyllene, present at 17.41%.

The *S. officinalis* specimens collected from the Adriatic coast of Croatia exhibited high levels of α-thujone, β-thujone, camphor, and 1,8-cineole [18]. In the essential oil of S. officinalis sourced from Tunisia, 49 different components were identified. The primary constituent was camphor, followed by a significant presence of α-thujone at 18.83%. Additionally, significant components included 1,8-cineole, viridiflorol, β-thujone, and β-caryophyllene [19]. In *S. officinalis* obtained from the eastern Adriatic coast flora and islands, this plant’s essential oil contains significant amounts of dominant compounds, specifically thujones and camphor [20]. The major components found in the essential oil of *O. compactum* collected in Morocco were camphor, carvacrol, thymol, and *p*-cymene [21]. Carvacrol, linalool, and *p*-cymene were identified as the primary constituents of the essential oil of *O. compactum* from Turkey [22]. Another study on the essential oils of this plant from China revealed that the main constituents were methyl eugenol, myristicin, carvacrol, and thymol [23]. An essential oil analysis of *clove* (*S. aromaticum*) grown in Mexico unveiled eugenol as the primary compound, accounting for a minimum of 50% of the oil. The remaining 10 to 40% consists of eugenyl acetate, β-caryophyllene, and α-humulene [24]. The main constituents of *S. aromaticum* grown in Algeria are eugenol, β-caryophyllene, and eugenyl acetate [25].

The significant variation in the essential oil composition observed between our studies and other studies can be attributed to various factors, including the conditions that influence the seasonal fluctuations of plant metabolites [26]. Regarding the genotype of the plant, the phenological stage of the plant at the time of harvest, environmental conditions, the plant parts used for essential oil extraction, and the drying method employed, all these factors can influence the chemical composition of the essential oil and account for the differences observed between studies [20]. Indeed, the variations in chemical composition observed in essential oils have given rise to chemotypes [27]. Chemotypes are typically described as distinct populations within the same species that exhibit variations in their chemical profiles, particularly for specific classes of secondary metabolites. These chemotypes can be identified based on the unique combination and abundance of critical compounds present in their essential oils.

**Table 1 plants-12-03376-t001:** Chemical composition of *O. compactum* (HO), *S. Officinal* (HS), and *S. aromaticum* (HC).

No.	Classification of Volatile Compounds	Compounds	Tr (min)	RI	RI (Literature) [28,29]	HS ^a^	HO ^b^	HC ^c^
1	**Monoterpene Hydrocarbons**	**α-thujene**	5.08	953	919	-	1.06	-
2		**α-pinene**	5.21	957	927	1.4	0.95	-
3		**camphene**	5.47	966	945	0.2	0.02	-
4		**β-pinene**	5.94	981	980	0.27	0.2	-
5		**β-myrcene**	6.11	986	991	0.31	1.73	-
6		**α-terpinene**	6.59	1022	1015	0.52	0.7	-
7		** *p* ** **-cymene**	6.74	1025	1026	17.77	13.33	0.12
8		**γ-terpinene**	7.3	1036	1062	-	9.98	-
9	**Oxygenated Monoterpenes**	**1,8-cineole**	8.29	1053	1028	0.41	0.09	-
10		**camphor**	8.79	1061	1137	0.61	2.89	-
11		**α-terpineol**	9.34	1069	1189	-	0.55	-
12		**geranyl acetate**	9.57	1073	-	0.03	2.25	-
13		**α-thujone**	10.84	1089	-	78.04	0.65	-
14		**thymol**	11.19	1092	1291	0.26	12.73	-
15		**carvacrol**	11.36	1174	-	0.18	37.68	-
16	**Phenylpropanoid**	**eugenol**	12.08	1281	-	-	11.22	72.66
17	**Sesquiterpene Hydrocarbons**	**α-copaene**	12.31	1457	-	-	-	0.02
18		**β-caryophyllene**	12.99	1485	-	-	2.28	17.41
19		**cis-calamenene**	14.37	1537	-	-	0.06	4.83
20	**Oxygenated Sesquiterpenes**	**caryophyllene oxide**	15.23	1566	1581	-	1.63	4.5

^a^ (*S. Officinal*), ^b^ (*O. compactum*), ^c^ (*S. aromaticum*); Tr: Retention time; RI: Retention index.

### 2.2. Antioxidant Activity

The ability of a chemical to prevent or reduce oxidative damage caused by free radicals in the body is referred to as its antioxidant potential. It measures the cumulative antioxidant activity present in the substance or sample. Depending on the individual characteristics and objectives of the study, various methods for measuring antioxidant potential can be used. In our work, we evaluated this activity using the following methods, the DPPH• radical scavenger activity and the TAC test.

The TAC of essential oils from the three oils was evaluated using the phosphomolybdenum test. This method relies on the reduction of Mo (VI) to Mo (V) by the various antioxidants present in oils, resulting in the formation of a green-colored phosphate/Mo (V) complex. During the test, it was observed that the absorbance increased proportionally with increasing concentrations (Figure 3). At a 100 mg/mL concentration, *O. compactum* (HO) essential oil exhibited the highest total antioxidant activity (1086.81 ± 0.32 µM AAE/mg). *S. officinalis* (HS) and *S. aromaticum* (HC) oils followed, with total antioxidant activities of 250.27 ± 0.47 µM AAE/mg and 942.58 ± 0.24 µM AAE/mg, respectively. Notably, the phosphomolybdenum assay had not been previously employed to determine the antioxidant activity of the oils under investigation.

Additionally, for the DPPH assay, at a concentration of 100 mg/mL (Table 2 and Figure 4), the oil of *O. compactum* (HO) exhibited the highest anti-radical activity (IC_50_ = 0.12 ± 0.02 mg/mL) in comparison with ascorbic acid as a reference (IC_50_ = 0.26 ± 0.24 mg/mL) followed by *S. aromaticum* (HC) (IC_50_ = 0.42 ± 0.76 mg/mL) and *S. officinalis* (HS) (IC_50_ = 0.46 ± 0.23 mg/mL) extracts.

Based on the data, the DPPH scavenging capacity indicates the ability of the essential oils from these three plant sources (*O. compactum*, *S. officinalis* L., and *S. aromaticum*) to neutralize free radicals, which are reactive molecules that can cause oxidative damage in cells and tissues. The oxygenated monoterpenes (carvacrol, thymol, and eugenol), the main components of the oils studied, are three compounds found in various plant sources. Because of their ability to scavenge free radicals and alleviate oxidative stress, these chemicals are recognized for their antioxidant characteristics and have been researched for their potential health benefits [30,31]. These essential oils’ scavenging activity indicates their potential health advantages and applications.

### 2.3. Antibacterial Activity

The chemical diversity of the molecules that make up essential oils defines their antibacterial power. This study aimed to demonstrate the antibacterial activity of essential oils extracted from HC, HO, and HS medicinal and aromatic plants against *Staphylococcus aureus*, *Listeria innocua*, *Escherichia coli*, and *Pseudomonas aeruginosa* bacteria. The chromatograms showed inhibition diameters between 8 and 10 mm for HO and HS oils and 10 and 13 mm for HC oils (Table 3). That means the three essential oils have low activity compared to gentamycin used as a reference, so there is no point in conducting the quantitative tests (calculation of MIC and CMF).

The antifungal activity was evaluated against the yeasts *Candida albicans*, *Candida glabrata*, *Saccharomyces cerevisiae*, *Aspergillus niger*, and *Geotrichum candidum*. Measurement of inhibition diameters gave poor results for essential oils HS and HO against all the fungal strains tested. However, the oil of HC gave good results against all tested strains (Table 4). The activity of the HC oil gave a zone of inhibition of 14.5 ± 0.2 against *C. albicans*, 21.5 ± 0.1 against *C. glabrata*, 16 ± 0.1 against *S. cerevisiae*, 22.5 ± 0.1 against *A. niger*, and 18 ± 0.2 against *G. candidum*. That means essential oil HC has potent antifungal activity against all the fungal strains tested. The results obtained for the antifungal activity of essential oil HC led us to move on to quantitative tests and thus measure this oil’s minimum inhibitory and fungicidal concentrations against all the fungal strains tested.

The lowest MIC of this oil was recorded against *A. niger*, which was equal to 0.125%, followed by 0.25% against *C. glabrata*, 0.5% against *G. candidum*, 1% against *S. cerevisiae*, and 2% against *Candida albicans* (Table 5), with MFCs ranging from 0.25% against *A. niger* to 8% against *S. cerevisiae* and *G. candidum*. These results confirm that this essential oil has a potent antifungal activity and can be used as an alternative against various diseases caused by these fungal types and even for treating phytopathology of a fungal origin. The bioactive compounds in HC essential oil could be responsible for this solid antifungal activity [32]. β-caryophyllene, one of plants’ major compounds, contributes significantly to their antifungal activity against various fungal strains. Carvacrol, another bioactive compound, is also responsible for the antifungal effects observed in the essential oils studied. It can disrupt fungal cell membranes, hindering their growth and dissemination. In addition, it can interfere with the mechanisms that regulate fungal growth, reducing their viability. These results suggest that these three essential oils could be used as natural alternatives in various fields, including food, health, and pharmaceutical research [31,33,34].

### 2.4. In Silico Prediction

The drug discovery process is a complex undertaking, and selecting the right lead molecule is paramount for the overall success of the project [35,36]. Molecular docking has been a widely used computational technique in structure-based drug design since the early 1980s [36,37,38]. The molecular docking process involves two main stages: firstly, predicting the shape, position, and orientation of the ligand (typically a small molecule) within the protein’s binding site, and secondly, evaluating the binding quality using a scoring function. Ideally, the sampling method should accurately reproduce the actual binding mode observed in experiments, and the scoring function should rank it as the most favorable among all generated binding positions [36].

In this particular study, researchers utilized molecular docking to investigate the potential underlying mechanism responsible for the antioxidant, antifungal, and antibacterial properties of three components found in essential oils. The researchers used binding affinity values, represented using ΔG, to assess how strongly the compounds preferred the target proteins compared to a known inhibitor. Specifically, they employed this method to examine the binding affinities of 20 compounds present in essential oils to two proteins associated with bactericidal/bacteriostatic activity: DNA gyrase topoisomerase II (PDB ID: 1KZN) and enoyl-acyl carrier protein reductase (PDB ID: 3GNS) [39,40,41,42]. Furthermore, they investigated two proteins related to antifungal activity, namely cytochrome P450 14 alpha-sterol demethylase (PDB ID: 1EA1) and N-myristoyl transferase (PDB ID: 1IYL) [39,40,43].

The results of the molecular docking experiments are presented using a heat map table, which employed a color gradient (Table 6). Topoisomerase enzymes have gained significant attention in the development of medications due to their crucial role in DNA replication [44]. Among them, DNA gyrase, a type II topoisomerase found in all bacteria, plays a vital role in determining DNA’s topological state. Releasing the supercoiled DNA is essential for processes such as replication and transcription. Consequently, many antibiotics and antimicrobial agents primarily function by inhibiting DNA gyrase and blocking this step [45]. Given the importance of DNA gyrase, our in silico research primarily focused on DNA gyrase topoisomerase II (PDB: 1KZN), an enzyme derived from the *E. coli* bacterium. Studies have demonstrated that DNA gyrase controls the topological structure of bacterial genomes and is present in all bacterial species [46,47].

The compounds from the essential oils, when docked to protein 1KZN (DNA gyrase topoisomerase II), exhibited moderate binding affinities ranging from −4.7 to −6.7 kcal/mol. However, they did not demonstrate significant inhibitory potential compared to the native ligand Clorobiocin, which achieved a high docking score of −9.6 kcal/mol. This stark difference in inhibitory potential can be attributed to the formation of two conventional hydrogen bonds between Clorobiocin and specific amino acid residues (THR A:165 and ASP A:73) on the protein [48]. These hydrogen bonds likely play a crucial role in stabilizing the complex and enhancing inhibitory activity. The structural diversity of the essential oil compounds, variations in chemical structures, and potentially suboptimal orientations may contribute to their lower inhibitory potential [48].

Bacteria utilize fatty acid biosynthesis to construct and maintain their cell membranes and other essential cellular structures. This process involves a series of enzyme-driven reactions that convert simple precursors into long-chain, unsaturated fatty acids, which form the majority of these structures. The final step in this pathway involves reducing the double bond in an intermediate called an enoyl-ACP derivative. This reduction step is catalyzed by an enzyme known as enoyl-acyl carrier protein (ACP) reductase or FabI [49].

FabI plays a crucial role in the efficiency and effectiveness of the fatty acid biosynthesis pathway. Inhibiting FabI can result in the accumulation of toxic intermediates and eventually cause cell death. Therefore, FabI is a promising target for developing new antibiotics [50,51]. FabI, particularly the main enoyl reductase enzyme, FabI, is found in various microorganisms, including bacterial strains. In line with previous investigations, this study selected the enoyl-acyl carrier protein reductase (*S. aureus*) protein (PDB ID: 3GNS) as a potential target. Scientists are currently focusing on FabI as a potential target for developing antibacterial compounds. FabI crystal structures have been identified in different types of bacteria, such as *E. coli* and *S. aureus* [52]. In the context of our study, β-caryophyllene and caryophyllene oxide from HO and HC emerged as robust inhibitors of the FabI protein. These compounds displayed binding affinities of −6.2 kcal/mol, mirroring the binding affinity of the native ligand Triclosan (−6.2 kcal/mol). Notably, Triclosan formed a single conventional hydrogen bond with ILE A:193 (Figure 5a). In comparison, our two compounds established a similar interaction, forming a conventional hydrogen bond with the amino acid residue ASN A:205 at the active site of FabI (Figure 5b). This structural similarity in hydrogen bonding interactions suggests that β-caryophyllene and caryophyllene oxide possess inhibitory potential against FabI comparable to that of the reference compound Triclosan.

The essential role of the enzyme cytochrome P450 14 α-sterol demethylase (CYP51s; PDB ID: 1EA1) is to facilitate the synthesis of sterols in fungi by catalyzing the production of intermediate compounds, particularly ergosterol [53]. CYP51s, recognized as a pivotal enzyme in sterol production, is a prime target for antifungal medications [54]. However, our study on this enzyme revealed that none of the tested compounds exhibited significant inhibitory effects on the fungal protein. The binding energies of these compounds ranged from −3.2 to −5.5 kcal/mol, indicating their lower potency when compared to the established antifungal drug Fluconazole, which demonstrated a binding energy of −5.8 kcal/mol. These results strongly suggest that the observed antifungal effects of the three essential oils are not attributed to the inhibition of this particular protein. It underscores the possibility that alternative mechanisms or target proteins may be responsible for the antifungal properties exhibited with the essential oils. In the research conducted by Kandsi et al. in 2022 [50], it was found that (+)-4-Carene, present in the essential oil of *D. ambrosioides*, exhibited a robust inhibitory impact on the fungal protein. This inhibition was characterized by a binding energy of −6.1 kcal/mol, surpassing the potency of the current antifungal medication Fluconazole, which displayed a binding energy of −5.8 kcal/mol.

The protein known as N-myristoyl transferase (NMT; PDB ID: 1IYL) is exclusively found in eukaryotic cells and serves as a catalyst for the attachment of myristate fatty acid to the N-terminal glycine of other proteins. It utilizes myristoyl-CoA as a substrate [55]. NMT (N-myristoyltransferase) is required for the growth of fungal infections, cell death, and signal transduction, among other biological activities [56]. In the case of α-copaene and cis-calamenene from HC, as well as β-caryophyllene and caryophyllene oxide from HO and HC, these compounds demonstrated inhibitory potential on NMT with binding affinities of −6.6, −6.5, −5.8, and −6.4 kcal/mol, respectively. These values were compared with the binding affinity of the potent antifungal agent Fluconazole (−5.8 kcal/mol). Upon analyzing the two-dimensional pattern of chemical interactions with the protein’s active site, we made intriguing observations regarding the mechanisms of action of certain compounds. For instance, Fluconazole, a reference antifungal agent, was found to create a solitary typical hydrogen bond with SER A:120 (Figure 6a). In contrast, the potent inhibitor α-copaene, identified in our study, did not establish any typical hydrogen bonds with the protein’s active pocket (Figure 6b). This divergence in interaction mechanisms suggests that α-copaene may employ a distinct inhibitory process, differing from the conventional hydrogen bond formation seen with Fluconazole.

These findings strongly imply that α-copaene and similar compounds may be responsible for the observed antifungal activity in HO and HC against the studied fungi. Their unique interaction patterns with the protein’s active site point to alternative mechanisms of inhibition, highlighting their potential as novel antifungal agents worthy of further investigation.

Lipoxygenases are a type of enzyme that utilize a redox mechanism to facilitate the oxidation of PUFAs. This enzymatic process leads to the production of a hydroperoxide, which is an oxygen-centered radical derived from the fatty acids. The presence of these radicals can potentially contribute to the onset and progression of various serious diseases [57,58]. Two specific proteins, lipoxygenase (1N8Q) and cyto-chrome P450 (1OG5), were selected for investigation. Regarding lipoxygenase, eight ligands, namely α-thujene, carvacrol, thymol, α-copaene, camphene, β-caryophyllene, cis-calamenene, and caryophyllene oxide, were identified as potent inhibitors. These ligands exhibited binding affinities of −6.5, −6.2, −6.2, −6.3, −6.1, −6.1, −6.2, and −6.6 kcal/mol, respectively. The binding energy value of −6.0 kcal/mol was observed for the native ligand protocatechuic acid, in contrast to the comparison compound. The 2D interactions of both protocatechuic acid and the potent ligand found in the essential oils, caryophyllene oxide, with the active site of lipoxygenase are presented in Figure 7a,b.

In the case of the second protein of interest, cytochrome P450 (PDB ID: 1OG5), cis-calamenene from HC, as well as caryophyllene and caryophyllene oxide from HO and HC, demonstrated inhibitory effects; cis-calamenene exhibited an inhibitory activity with a binding energy of −7.3 kcal/mol, while caryophyllene and caryophyllene oxide showed binding energies of −7.4 and −7.3 kcal/mol, respectively. These values were compared to the free binding energy of −6.6 kcal/mol, which corresponds to the native inhibitor warfarin [59]. Figure 8a,b depict the two-dimensional interactions between warfarin and the powerful ligand derived from essential oils, caryophyllene, and the active site of NADPH oxidase (PDB ID: 1OG5).

## 3. Materials and Methods

### 3.1. Plant Materials

The aerial part of each of the studied plants (*S. officinalis*, *O. compactum*, and *S. aromaticum*) was gathered from Taourirt (Northeastern Morocco, coordinates 34°8′31″ N, 2°59′52″ W) in March 2023. They were located in the Oriental region, approximately 100 km west of Oujda, as described in Figure 9, and were chosen to further this work.

Fresh species were consecutively gathered in sterile plastic bags, transferred to the laboratory, and chopped into small fragments.

### 3.2. Hydrodistillation Protocol

The essential oils of three species (*S. officinalis*, *O. compactum*, and *S. aromaticum*) (Figure 10) were extracted with hydrodistillation using Clevenger-apparatus equipment, as described by Azghar, A. et al. [60]. The three fresh species (150 g) were accurately weighed and combined with 500 ml of purified water. They were subjected for 2 h to 100 °C until the EO level stabilized. Essential oils were collected, dried over anhydrous Na_2_SO_4_, and stored in sealed amber vials (4–6 °C) until the analysis. The essential oil hydrodistillation yield was calculated as the oil mass after distillation divided by the mass of fresh organs.

### 3.3. GC-MS Analysis

The phytochemical study of EO was characterized and identified using a Shimadzu GC system (Kyoto, Japan) equipped with a BPX25 capillary column with a 5% diphenyl and 95% dimethylpolysiloxane phase (30 m, 0.25 mm, and 0.25 m) and coupled to a QP2010 MS. The mobile phase was helium gas (99.99%) at a flow rate of 1.69 L/min. The injection, ion source, and interface temperatures were set to 250 °C, while the temperature program for the column oven was set to 50 °C for 1 min before being heated to 250 °C at a rate of 10 °C/min and maintained for an additional minute. The sample components were ionized in the Electron Ionization (EI) mode at 70 eV and a scanned mass range of 40 to 300 *m*/*z*. Each extract was administered in a 1 µL volume in the split mode. The compounds were identified by comparing their retention durations, mass spectra fragmentation patterns, and databases such as the National Institute of Standards and Technology’s (NIST) database. LabSolutions (version 2.5, Shimadzu, Kyoto, Japan) was used to process the data [61,62,63].

### 3.4. DPPH Scavenging Assay

The test was evaluated using the in vitro DPPH assay, as described in [32], with minor adjustments. The substance’s essential oil was diluted in a series of concentrations (5 to 200 mg/mL were created) to determine the antioxidant properties. Then, a 1 mL volume of each dilution was combined with 2.5 mL of a 0.1 mM methanolic DPPH solution. The mixture was incubated in darkness for 30 min, measuring at 517 nm. Each experiment was repeated thrice, and the scavenging activity was calculated using the provided formula [58,64,65].
$$\mathrm{Free}\;\mathrm{Radical}\;\mathrm{Scavenging}\left(\%\right)=\left[\left(\frac{A_{blank}-A_{sample}}{A_{blank}}\right)\right]\times 100$$

The measurements were taken for the absorbance in the absence of the sample (A_0_) and the absorbance in the presence of the sample (A_1_). Ascorbic acid was employed as the reference compound.

The amount of DPPH inhibited using the essential oils was expressed as the percent concentration corresponding to a 50% inhibition (IC_50_).

### 3.5. Total Antioxidant Capacity

The total antioxidant capacity was determined using the phosphor-molybdenum method [58]. The sample extract or standard solution was combined with a 0.6 M sulfuric acid reagent solution, 28 mM of sodium phosphate, and 4 mM of ammonium molybdate. Subsequently, the mixture was incubated at 95 °C for 90 min. The resulting solution’s absorbance was measured at 695 nm. The antioxidant capacity was expressed as ascorbic acid equivalents, determined using a standard curve established with ascorbic acid [66]. The blank solution, which did not contain the test sample, included all the reagents. The experiments were conducted three times.

### 3.6. Choice of Strains and Agar Diffusion Method

The in vitro antibacterial activity of the samples was assessed with the suitable diffusion method on Mueller–Hinton agar (MHA) against two Gram-negative bacteria, *Escherichia coli* and *Pseudomonas aeruginosa* (ATCC 49,189), and two Gram-positive bacteria, *Staphylococcus aureus* (ATCC 6538) and *Listeria monocytogenes* (ATCC 19,117), and antifungal activity was tested against *Geotrichum candidum*, *Aspergillus niger*, *Saccharomyces cerevisiae*, *Candida glabrata*, and *Candida albicans* on PDA agar. Briefly, agar Petri plates were inoculated with bacterial strains. Sterilized filter paper discs (6 mm in diameter) were filled with 20 µL of the essential oils. Gentamicin (1 mg/mL) was used as a positive control and DMSO as a negative control. The discs containing the samples, the standard antibiotic (gentamicin), and the DMSO were deposited on the agar surface using flamed forceps, then slightly compressed to ensure good contact between the discs and the surface. After pre-diffusing for 30 min at room temperature, the Petri dishes were incubated at 37 °C for 18 h. The diameters of the inhibited growth zones were then measured in millimeters (mm) [67].

### 3.7. Determination of MIC in Liquid Medium

The minimum inhibitory concentration (MIC) is the lowest concentration that effectively inhibits bacterial growth after a 24 h incubation period. The MIC value was determined by visually observing the presence or absence of a red color indicator [43,68].

### 3.8. Determination of the MFC on Solid Media

The Minimum Fungicidal Concentration (MFC) corresponds to the lowest essential oil concentration capable of killing more than 99.9% of the initial fungal inoculum. The MFC was determined for yeasts by taking 3 µL from the negative wells, depositing them on the solid YEG medium, and then incubating it at 25 °C for 48 h. The same protocol was used for molds, except that incubation was carried out at 25 °C for 72 h. After incubation, the CMF corresponds to the lowest essential oil concentration, showing no growth [43,69].

### 3.9. Molecular Docking Procedure

We used molecular docking approaches to estimate the antioxidant, antibacterial, and antifungal activities of the discovered compounds contained in three essential oils to evaluate their potential therapeutic effects. The approach used in this investigation adhered to a pre-established protocol specified in earlier research [50,58,70]. The automated docking experiments were performed using Auto Dock Vina v1.5.6 software [71]. The study focused on two antibacterial proteins, DNA gyrase topoisomerase II (PDB ID: 1KZN) [72,73] and enoyl-acyl carrier protein reductase (PDB ID: 3GNS) [41,42], as well as two antifungal target proteins, cytochrome P450 14 α-sterol demethylase (PDB ID: 1EA1) and N-myristoyl transferase (PDB ID: 1IYL) [4,5,74,75]. Additionally, two protein structures, lipoxygenase and CYP2C9 (PDB IDs: 1N8Q and 1OG5, respectively), were selected as antioxidant proteins [59,76,77].

## 4. Conclusions

The current research adds to our understanding of the phytochemical and biological activities of Moroccan *O. compactum* (HO), *S. officinalis* (HS), and *S. aromaticum* (HC) essential oils generated using HDs. The results suggest that the primary ones in *S. officinalis* essential oil are α-thujone and *p*-cymene, accounting for 78.04% and 17.77%, respectively. Thymol was the most abundant compound found in *O. compactum*, accounting for 37.68% of the total, followed by carvacrol (12.73%) and *p*-cymene (13.33%). The discovered primary components of *S. aromaticum* essential oil were eugenol (72.66% concentration) and caryophyllene (17.41% concentration). We discovered that HO and HS have low antibacterial activity except for HC, which has highly remarkable antifungal activity. Furthermore, the findings suggested that HO oil has significant antioxidant effects. The presence of α-thujene, carvacrol, thymol, α-copaene, camphene, β-caryophyllene, cis-calamenene, and caryophyllene oxide was found as a significant inhibitor of bioactive chemicals based on the results of the *in silico* assays, which were compatible with the experimental results. In conclusion, the study indicated that these natural metabolites in essential oils of *O. compactum* (HO), *S. officinalis* (HS), and *S. aromaticum* (HC) could be good alternatives to standard approved drugs and can support the usefulness of HC, HS, and HO in nutritional and medicinal contexts.

## Figures and Tables

**Figure 1 plants-12-03376-f001:**
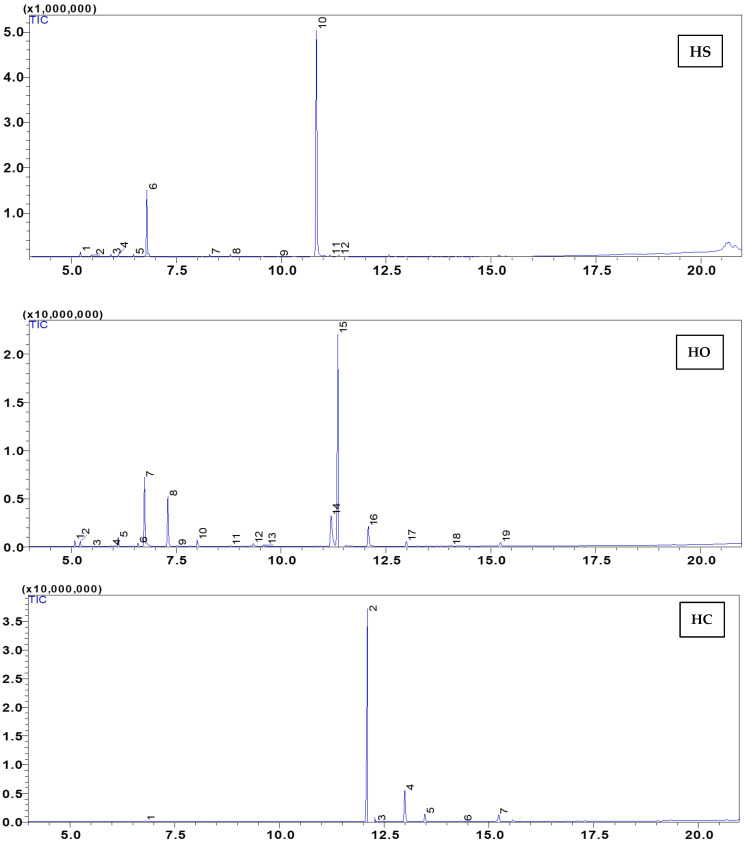
GC-MS chromatogram of *O. compactum* (HO), *S. officinalis* (HS), and *S. aromaticum* (HC).

**Figure 2 plants-12-03376-f002:**
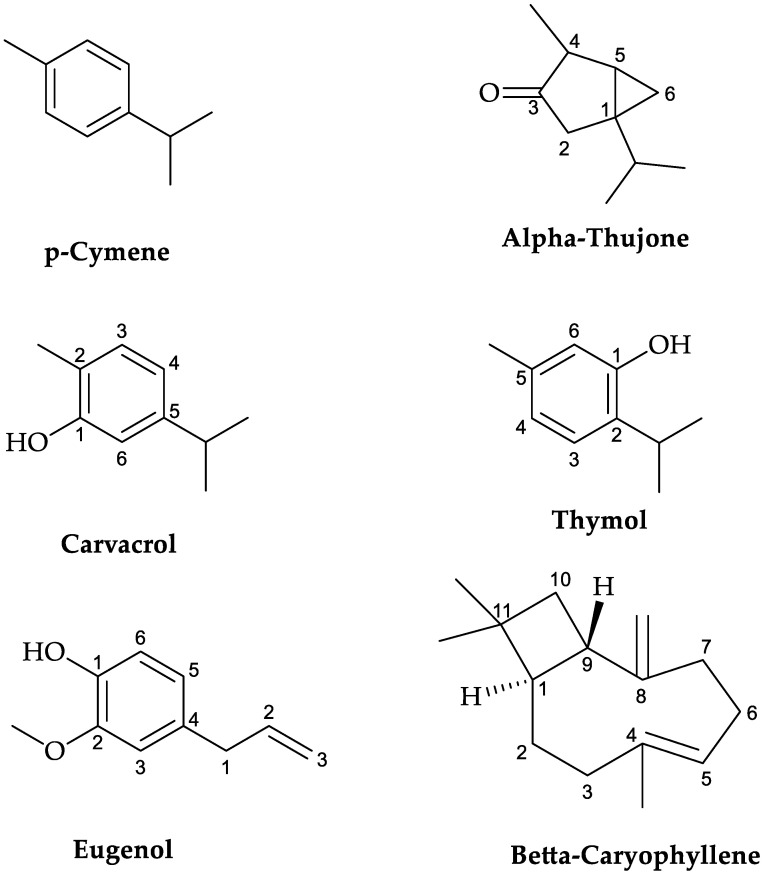
Structural formulas of the main components of *O. compactum* (HO), *S. officinalis* (HS), and *S. aromaticum* (HC) essential oils.

**Figure 3 plants-12-03376-f003:**
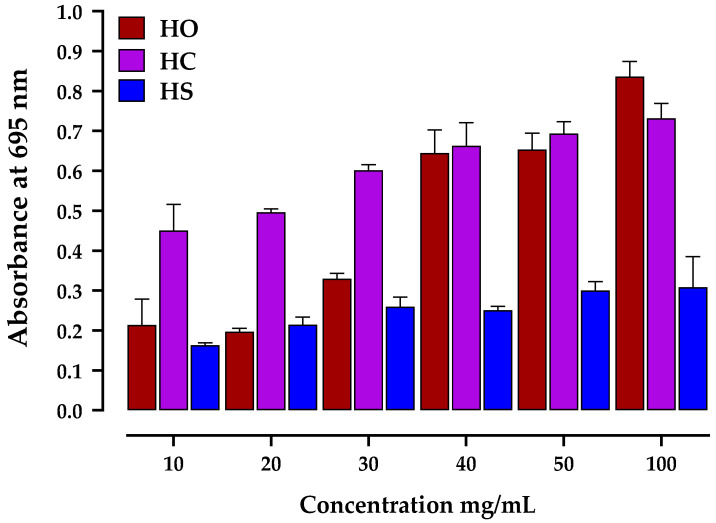
Effect of essential oil concentration on *O. compactum* (HO), *S. Officinalis* (HS), and *S. aromaticum* (HC) total antioxidant activity (phosphomolybdate assay). The mean SD of three determinations is used to calculate the values.

**Figure 4 plants-12-03376-f004:**
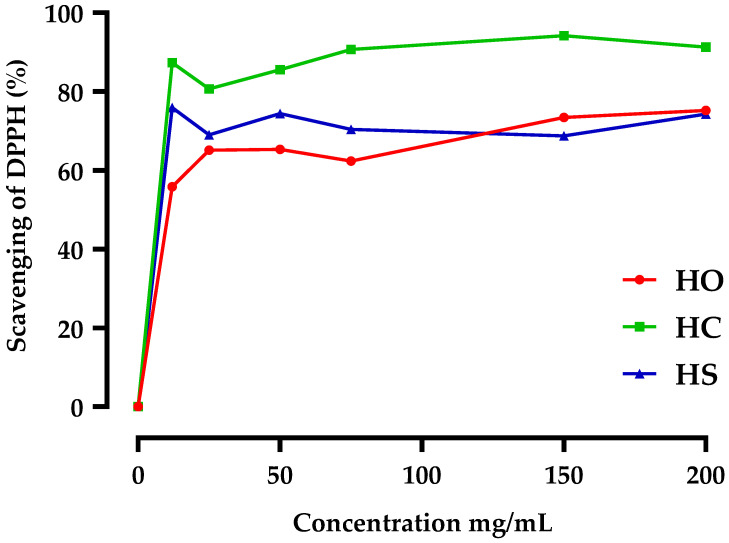
Effect of essential oil concentration on *O. compactum* (HO), *S. officinal* L. (HS), and *S. aromaticum* (HC) (DPPH scavenging capacity). The mean SD of three determinations is used to calculate the values.

**Figure 5 plants-12-03376-f005:**
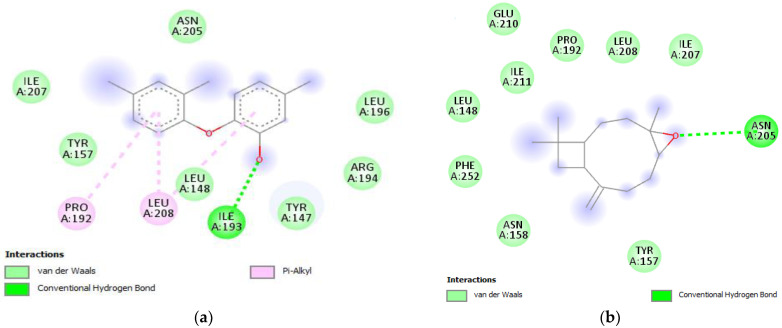
Two-dimensional presentation of the interaction of the native ligand Triclosan (**a**), and β-caryophyllene oxide (**b**), with the active site of FabI (PDB ID: 3GNS).

**Figure 6 plants-12-03376-f006:**
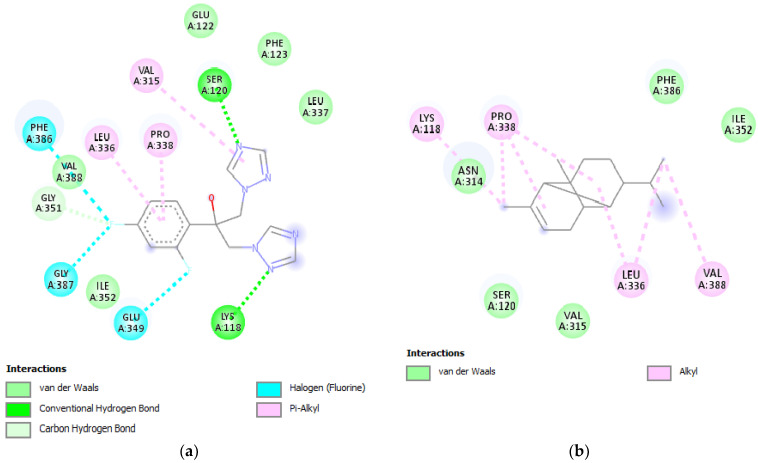
Two-dimensional presentation of the interaction of the antifungal agent Fluconazole (**a**), and α-copaene (**b**), with the active site of NMT (PDB ID: 1IYL).

**Figure 7 plants-12-03376-f007:**
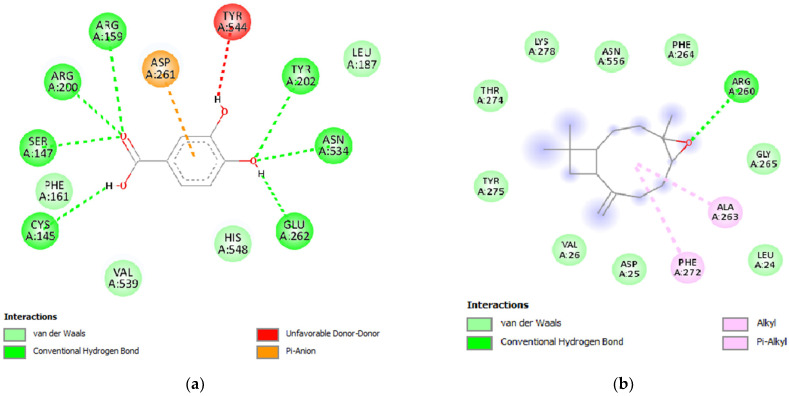
Two-dimensional interactions of the native ligand protocatechuic acid (**a**), and caryophyllene oxide (**b**), with the active site of lipoxygenase (PDB ID: 1N8Q).

**Figure 8 plants-12-03376-f008:**
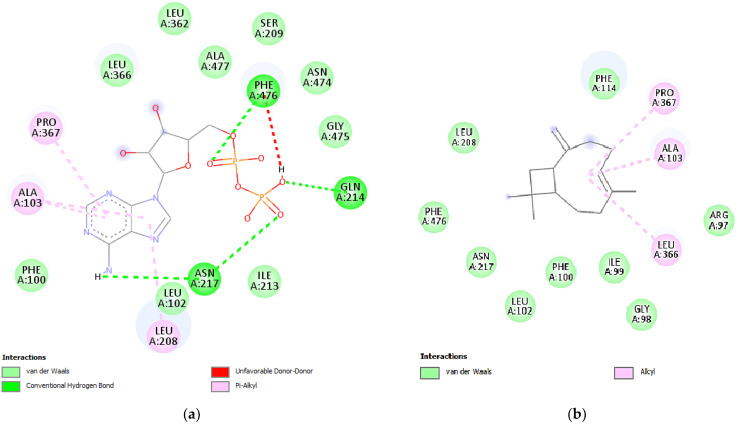
Two-dimensional interactions between the native ligand protocatechuic acid (**a**) and β-caryophyllene (**b**) with the active site of lipoxygenase (PDB ID: 1N8Q).

**Figure 9 plants-12-03376-f009:**
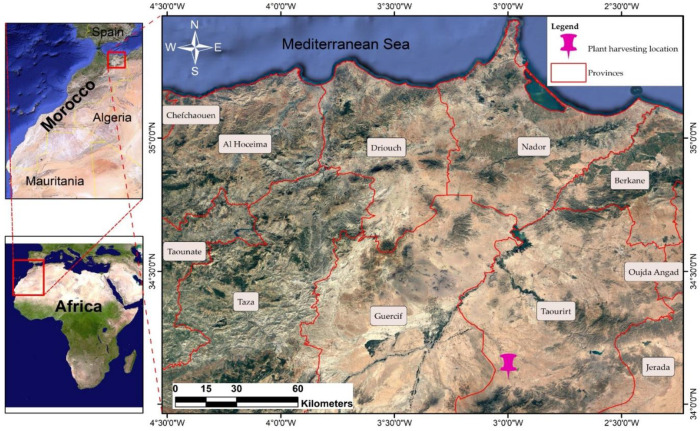
Map of the study area, showing the harvesting location.

**Figure 10 plants-12-03376-f010:**
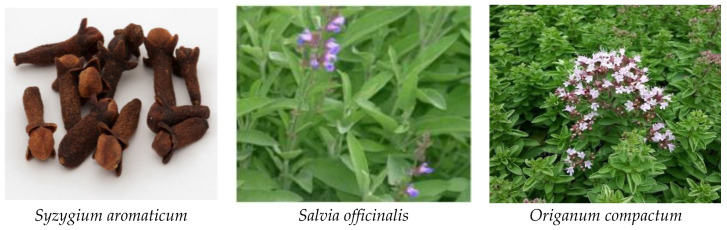
The three species (*S. officinalis*, *O. compactum*, and *S. aromaticum)* used in this study.

**Table 2 plants-12-03376-t002:** The antioxidant and free radical scavenging ability of *O. compactum* (HO), *S. Officinal* L. (HS), and *S. Aromaticum* (HC).

Oils/Standard	DPPH Scavenging Capacity IC_50_ (mg/mL)	Total Antioxidant Capacity *
**HO**	0.12 ± 0.02	1086.81 ± 0.32
**HC**	0.42 ± 0.76	942.58 ± 0.24
**HS**	0.46 ± 0.23	250.27 ± 0.47
**IC_50_ (Ascorbic acid (AA))**	0.26 ± 0.24	-

* Total antioxidant capacity expressed in µM ascorbic acid equivalents/mg oil.

**Table 3 plants-12-03376-t003:** Diameters (mm) of the bacterial growth inhibition zones induced using one of three species (*S. officinalis*, *O. compactum*, and *S. Aromaticum* (20 μg/disque)), as well as gentamycine (1 mg/mL/disc).

Oils/Antibiotic	IZ (mm) ^a^			
	*E. coli*	*L. innocua*	*S. aureus*	*P. aeruginosa*
**HS ^b^**	9.00 ± 0.31	8.00 ± 0.22	10.00 ± 0.26	10.00 ± 0.10
**HO ^c^**	8.00 ± 0.12	8.00 ± 0.43	10.00 ± 0.25	10.00 ± 0.22
**HC ^d^**	10.00± 0.23	10.00 ± 0.14	11.00 ± 0.22	13.00 ± 0.23
**Gentamycine ^e^**	29.50 ± 0.60	29.90 ± 0.20	30.10 ± 0.40	21.30 ± 0.11

^a^ Inhibition zone; ^b^
*S. officinal* L.; ^c^
*O. compactum*; ^d^
*S. aromaticum*; ^e^ Positive control.

**Table 4 plants-12-03376-t004:** HS, HO, and HC essential oils’ effect on bacterial strains.

Oils/Antibiotic	IZ (mm) ^a^				
	*C. albicans*	*C. glabrata*	*S. cerevisiae*	*A. niger*	*G. candidum*
**HS ^b^**	8.00 ± 0.21	7.00 ± 0.20	-	-	-
**HO ^c^**	9.00 ± 0.27	10.00 ± 0.35	7.00 ± 0.23	13.50 ± 0.28	11.00 ± 0.12
**HC ^d^**	14.50 ± 0.23	21.50 ± 0.17	16.00 ± 0.13	22.50 ± 0.14	18.00 ± 0.20
**Gentamycine ^e^**	21.40 ± 0.13	41.32 ± 0.03	67.50 ± 0.29	48.12 ± 0.01	23.00 ± 0.28

^a^ Inhibition zone; ^b^
*S. officinal* L.; ^c^
*O. compactum*; ^d^
*S. aromaticum*; ^e^ Positive control.

**Table 5 plants-12-03376-t005:** Minimum inhibitory (MIC) and fungicidal concentrations (MFC) of HC (*S. aromaticum*) against the studied fungal strains.

Oils/Antibiotic	*C. albicans*	*C. glabrata*	*S. cerevisiae*	*A. niger*	*G. candidum*
**MIC (%)**	2	0.25	1	0.125	0.5
**MFC (%)**	8	1	8	0.25	4
**MFC/MIC**	4	4	8	2	8

**Table 6 plants-12-03376-t006:** Heat map of the docking scores (affinity values are expressed in kcal/mol) of the three essential oils’ bioactive components. The essential oils are HS, *S. officinalis*; HO, *O. compactum*; and HC, *S. aromaticum*; (+) present; (−) absent.

No.	Compounds	Compound Abundance in			Antibacterial Proteins (PDB IDs)		Antifungal Proteins (PDB IDs)		Antioxidant Proteins (PDB IDs)	
		HS	HO	HG	1KZN	3GNS	1EA1	1IYL	1N8Q	1OG5
					Free Binding Energy (Kcal/mol) *					
**-**	**native ligand**	**−**	**−**	**−**	**−9.6**	**−6.2**	**−5.8**	**−5.8**	**−6**	**−6.6**
**1**	**α-thujene**	**−**	+	**−**	−5.3	−4.7	−4.4	−4.5	**−6.5**	−5.6
**2**	**α-pinene**	+	+	**−**	−4.7	−5	−4.2	−4.9	−5.6	−5.6
**3**	**camphene**	+	+	**−**	−6.3	−5.1	−4.7	−4.8	**−6.1**	−6
**4**	**β-pinene**	+	+	**−**	−4.7	−4.9	−4.1	−4.8	−5.5	−5.6
**5**	**β-myrcene**	+	+	**−**	−4.9	−4.3	−3.2	−4.6	−4.1	−5.3
**6**	**α-terpinene**	+	+	**−**	−5.8	−4.9	−4.2	−4.7	−5.1	−6.1
**7**	***p*-cymene**	+	+	+	−6	−4.9	−4.2	−4.8	−4.1	−5.9
**8**	**γ-terpinene**	**−**	+	**−**	−5.8	−4.7	−4.3	−4.7	−5.1	−6.1
**9**	**1.8-cineole**	+	+	**−**	−5.3	−5.4	−4.3	−4.8	−5.4	−5.6
**10**	**camphor**	+	+	**−**	−5.5	−5.5	−4.4	−4.7	−5.5	−5.9
**11**	**α-terpineol**	+	+	**−**	−5.9	−5	−4.4	−5.1	−5.5	−5.7
**12**	**geranyl acetate**	+	+	**−**	−5.4	−4.8	−4.3	−4.8	−5	−6
**13**	**α-thujone**	+	−	**−**	−5.9	−4.9	−4.2	−4.6	−5.6	−6.1
**14**	**thymol**	+	+	**−**	−6.2	−5.1	−4.7	−4.7	**−6.2**	−6
**15**	**carvacrol**	+	+	**−**	−6	−5.4	−4.7	−5.3	**−6.2**	−6.3
**16**	**eugenyl**	**−**	+	+	−5.8	−5.3	−4.1	−4.8	−5.5	−5.5
**17**	**α-copaene**	**−**	−	+	−6.7	−5.6	**−5.2**	**−6.6**	**−6.3**	−5.6
**18**	**β-caryophyllene**	**−**	+	+	−5.3	**−6.2**	**−5.5**	**−5.8**	**−6.1**	**−7.4**
**19**	**cis-calamenene**	**−**	−	+	−6.1	−5.9	**−5.4**	**−6.5**	**−6.2**	**−7.3**
**20**	**caryophyllene oxide**	**−**	+	+	−5.6	**−6.2**	**−5.2**	**−6.4**	**−6.6**	**−7.3**

* The color scale for each column runs from red (representing the native ligand ∆G) through yellow (the midpoint) to green (representing the native ligand ∆G +4 kcal/mol). 1KZN: DNA Gyrase Topoisomerase II; 3GNS: Enoyl-Acyl Carrier Protein Reductase; 1EA1: Cytochrome P450 14 Alpha-sterol Demethylase; 1IYL: N-Myristoyl Transferase; 1N8Q: Lipoxygenase; 1OG5: CYP2C9.

## Data Availability

Data are contained within the article.

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
