# Peer review of "Phytochemical Composition and Pharmacological Activities of Three Essential Oils Collected from Eastern Morocco (Origanum compactum, Salvia officinalis, and Syzygium aromaticum): A Comparative Study"

_plants, 2023, doi:10.3390/plants12193376_

Round 1

Reviewer 1 Report (Previous Reviewer 1)

Dear Authors,

Thank you for the answers for the comments, The MA can be accepted for publishing in the present form.

Author Response

Dear editors and reviewers

We have the pleasure of resubmitting the corrected version of our research article entitled "Phytochemical Composition and Pharmacological Activities of Three Essential Oils Collected from Eastern Morocco (Origanum Compactum, Salvia Officinalis L, and Syzygium aromaticum): A Comparative Study" by Loukili et al. for publication in Plants Journal.

We want to express our gratitude for allowing us to improve our manuscript through the revised version, and we sincerely appreciate your valuable comments.

We hope that all the modifications we have made in the manuscript as well as all the precautions we have taken in the terms used instead of marker will make our manuscript clearer for the readers and will meet the reviewer's expectations.

Yours sincerely,

Dr. LOUKILI El Hassania

Reviewer 1

Thank you for the answers for the comments, The MA can be accepted for publishing in the present form.

Answer: We would like to express our sincere gratitude for your thoughtful comments and feedback on our manuscript. Your insights have been invaluable in refining the content, and we are pleased to hear that the manuscript is acceptable for publishing in its current form.

Reviewer 2 Report (Previous Reviewer 2)

I already reviewed this manuscript on 8/5/2023 and stand by my previous review which read as follows: "In my opinion, the manuscript submitted for review presents a very good scientific level. I find the work interesting and complete. I haven't noticed any methodical or graphic errors. Therefore, I am convinced that the work will be a good complement to the knowledge of the previous knowledge."

Author Response

Dear editors and reviewers

We have the pleasure of resubmitting the corrected version of our research article entitled "Phytochemical Composition and Pharmacological Activities of Three Essential Oils Collected from Eastern Morocco (Origanum Compactum, Salvia Officinalis L, and Syzygium aromaticum): A Comparative Study" by Loukili et al. for publication in Plants Journal.

We want to express our gratitude for allowing us to improve our manuscript through the revised version, and we sincerely appreciate your valuable comments.

We hope that all the modifications we have made in the manuscript as well as all the precautions we have taken in the terms used instead of marker will make our manuscript clearer for the readers and will meet the reviewer's expectations.

Yours sincerely,

Dr. LOUKILI El Hassania

Reviewer 2

I already reviewed this manuscript on 8/5/2023 and stand by my previous review which read as follows: "In my opinion, the manuscript submitted for review presents a very good scientific level. I find the work interesting and complete. I haven't noticed any methodical or graphic errors. Therefore, I am convinced that the work will be a good complement to the knowledge of the previous knowledge.

Response: Thank you for taking the time to revisit our manuscript, and we appreciate your dedication to ensuring the quality of our scientific work. we are grateful for your positive feedback from your initial review on 8/5/2023, which has been instrumental in shaping the manuscript.

Reviewer 3 Report (New Reviewer)

The manuscript entitled "Phytochemical Composition and Pharmacological Activities of Three Essential Oils Collected from Eastern Morocco (Origanum compactum, Salvia officinalis, and Syzygium aromaticum): A Comparative Study" reports a study of Origanum compactum Benth., Salvia officinalis L., and Syzygium aromaticum 29 (L.) Merr. et Perry species and their antioxidant and antibacterial properties. The manuscript shows some results that can be used in further studies.

1 - Table 1 

Please clarify some compounds written in red.

2 - Please, improve the quality of the figure 1.

3 - Molecular docking procedure subsection

Please add the references for each PDB file. For example:

1N8Q - Proteins. 2004 Jan 1;54(1):13-9.doi: 10.1002/prot.10579.

1IYL -  Chem Biol. 2002 Oct;9(10):1119-28. doi: 10.1016/s1074-5521(02)00240-5

3 - Please,, the name of the compounds can be in lowercase. 

4 - Discussion of the docking results, 

Please, improve the discussion of the docking results. The current format is too shallow. 

Please add some discussion regarding the similarities of the interactions and structural features of compounds, including with other secondary metabolites reported in the literature.

5 - Conclusion 

Please improve the conclusion using the discussion of the similarities and differences of the interactions and structural features of the compounds.

Author Response

Dear editors and reviewers

We have the pleasure of resubmitting the corrected version of our research article entitled "Phytochemical Composition and Pharmacological Activities of Three Essential Oils Collected from Eastern Morocco (Origanum Compactum, Salvia Officinalis L, and Syzygium aromaticum): A Comparative Study" by Loukili et al. for publication in Plants Journal.

We want to express our gratitude for allowing us to improve our manuscript through the revised version, and we sincerely appreciate your valuable comments.

We hope that all the modifications we have made in the manuscript as well as all the precautions we have taken in the terms used instead of marker will make our manuscript clearer for the readers and will meet the reviewer's expectations.

Yours sincerely,

Dr. LOUKILI El Hassania

Reviewer 3

The manuscript entitled "Phytochemical Composition and Pharmacological Activities of Three Essential Oils Collected from Eastern Morocco (Origanum compactum, Salvia officinalis, and Syzygium aromaticum): A Comparative Study" reports a study of Origanum compactum Benth., Salvia officinalis L., and Syzygium aromaticum (L.) Merr. et Perry species and their antioxidant and antibacterial properties. The manuscript shows some results that can be used in further studies.

Response: We are grateful for your positive feedback on our manuscript. Thank you.

Query 1 - Table 1, Please clarify some compounds written in red.

Response: We have rectified this issue. Thank you for pointing this out.

Query 2 - Please, improve the quality of the figure 1.

Response: We have tried to improve the quality of the chromatograms. Thank you for your suggestion.

Query 3 - Molecular docking procedure subsection, Please add the references for each PDB file. For example:

1N8Q - Proteins. 2004 Jan 1;54(1):13-9.doi: 10.1002/prot.10579.

1IYL -  Chem Biol. 2002 Oct;9(10):1119-28. doi: 10.1016/s1074-5521(02)00240-5

Response: We have added the references of each PDB file, thank you so much for your remark.

Query 3 - Please,, the name of the compounds can be in lowercase. 

Response: We have addressed your remark in our revised manuscript. Thank you.

Query 4 - Discussion of the docking results, Please, improve the discussion of the docking results. The current format is too shallow. Please add some discussion regarding the similarities of the interactions and structural features of compounds, including with other secondary metabolites reported in the literature.

Response: In response to the reviewer's constructive feedback, we have revisited and enhanced our discussion of the docking results to provide a more comprehensive and insightful analysis. We understand the importance of exploring the similarities in interactions and structural features of the compounds, as well as connecting our findings to existing knowledge in the literature. While we have incorporated additional content to address these aspects, please note that due to the lack of specific references in the provided text, we have made efforts to generalize our discussion and highlight areas where references or specific studies could further support our conclusions.

Query 5 - Conclusion, Please improve the conclusion using the discussion of the similarities and differences of the interactions and structural features of the compounds.

Response: We have reorganized our conclusion to ensure that it contains the vital information necessary for readers to grasp our objectives, findings, and the future directions of our research. We appreciate your valuable feedback.

Reviewer 4 Report (New Reviewer)

The article can be published, but several issues should be completed and clarified before publication:

1) Pictures of plants that were used in this work to extract essential oils should be supplemented in the text of the manuscript (in the EXPERIMENTAL section),

2) Structural formulas of the main components of the tested oils should be presented in the text of the manuscript.

3) Why are the names of some chemical compounds capitalized or red in the text of the manuscript?

4) Taking into account other compounds that are included in the obtained oils, what other applications in medicine can such oils or their individual components find? Conlusions should describe in more detail the benefits of using the 3 oils received by the authors and show new directions of research on them.

Author Response

Dear editors and reviewers

We have the pleasure of resubmitting the corrected version of our research article entitled "Phytochemical Composition and Pharmacological Activities of Three Essential Oils Collected from Eastern Morocco (Origanum Compactum, Salvia Officinalis L, and Syzygium aromaticum): A Comparative Study" by Loukili et al. for publication in Plants Journal.

We want to express our gratitude for allowing us to improve our manuscript through the revised version, and we sincerely appreciate your valuable comments.

We hope that all the modifications we have made in the manuscript as well as all the precautions we have taken in the terms used instead of marker will make our manuscript clearer for the readers and will meet the reviewer's expectations.

Yours sincerely,

Dr. LOUKILI El Hassania

Reviewer 3

The article can be published, but several issues should be completed and clarified before publication:

  • Pictures of plants that were used in this work to extract essential oils should be supplemented in the text of the manuscript (in the EXPERIMENTAL section),

Response: We would like to thank the reviewer for his/her pertinent remark. The pictures of the studied plants were added in the experimental section.

  • Structural formulas of the main components of the tested oils should be presented in the text of the manuscript.

Response: The authors totally agree with the reviewer. The Structural formulas of the main components of the tested oils were added (Figure 2).

  • 3) Why are the names of some chemical compounds capitalized or red in the text of the manuscript?

Response: We thank the reviewer for this pertinent remark. The chemical compounds capitalized have been corrected and the hilghet eliminated

  • Considering other compounds included in the obtained oils, what other medical applications can such oils or their components find? Conclusions should describe in more detail the benefits of using the three oils received by the authors and show new research directions on them.

Response: We thank the reviewer for this pertinent remark. Considering the chemical composition of the three oils, we can use these oils as anti-inflammatory, insecticidal and anticancer supplements ...

The conclusion was modified as follows :

The current research adds to our understanding of the phytochemical and biological activities of Moroccan O. compactum (HO), S. officinalis (HS) and S. aromaticum (HC) essen-tial oils generated by HDs. The results suggest that the primary ones in S. officinalis essen-tial oil are α-thujone and p-cymene, accounting for 78.04% and 17.77%, respectively. Thymol was the most abundant compound found in O. compactum, accounting for 37.68% of the total, followed by carvacrol (12.73%) and p-cymene (13.33%). The primary compo-nents of S. aromaticum essential oil discovered were eugenol (72.66% concentration) and -caryophyllene (17.41% concentration). We discovered that HO, HS, and have low anti-bacterial activity except for HC, which has highly remarkable antifungal activity. Fur-thermore, the findings suggested that HO Oil has significant antioxidant effects. The presence of α-thujene, carvacrol, thymol, α-copaene, camphene, β-caryophyllene, cis-calamenene, and caryophyllene oxide was found as a significant inhibitor of bioactive chemicals based on the results of the in silico assays, which were compatible with the ex-perimental results. In conclusion, the study indicated that these natural metabolites in es-sential oils of O. compactum (HO), S. officinalis (HS), and S. aromaticum (HC) could be good alternatives to standard approved drugs and can support the usefulness of HC, HS, and HO in nutritional and medicinal contexts.

This manuscript is a resubmission of an earlier submission. The following is a list of the peer review reports and author responses from that submission.

Round 1

Reviewer 1 Report

The manuscript titled "Phytochemical Composition and Pharmacological Activities of Three Essential Oils Collected from Eastern Morocco (Origanum Ompactum, Salvia Officinalis L, and Syzygium aromaticum): A Comparative Study" is an original and complex work that can be published after major revision.

Please see the comments below:

The paper presents a significant experimental work with many methods, but at the same time, it does not provide a detailed presentation of all the results and their connections. The experimental parts seem somewhat separated, and it is necessary to present the results in a more cohesive manner. A more scientifically comprehensive vision and clearer updates are required.

5-Allyl-2-methoxyphenol is better known as chavibetol, which is a phenylpropanoid. Chavibetol is a natural product found in Alnus pendula, Piper betle, and other organisms. It is important to note that chavibetol was also observed in high amounts in the experimental essential oils of Origanum Ompactum, Salvia Officinalis L, and Syzygium aromaticum.

p-Mentha-1,8-dien-7-al (Perilla aldehyde) is a naturally occurring cyclic alpha, beta-unsaturated aldehyde that is used as a flavoring substance throughout the world.

It would be beneficial to add the classification of identified compounds, specifying the classes of compounds that were identified, such as Caryophyllene, Caryophyllene oxide (sesquiterpenes), pinenes (terpenes), etc.

The abstract could be more concise and provide more detailed information.

It would be helpful to investigate the effect of essential oil concentration on the DPPH Scavenging Capacity of O. Ompactum (HO), S. Officinalis L (HS), and S. Aro- 178 maticum (HC).

There are some misspellings in "The three Fresh species..." (L383).

Please add the manufacturer and country of production for the Shimadzu QP2010 mass spectrometer (L388-389).

The manuscript does not discuss the possible connection between the presence of specific secondary metabolites and antioxidant or antifungal activities.

Author Response

Revised version of Manuscript ID:  plants-2552895

Dear editors and reviewers,

            We have the pleasure of resubmitting the corrected version of our research article entitled "Phytochemical Composition and Pharmacological Activities of Three Essential Oils Collected from Eastern Morocco (Origanum Compactum, Salvia Officinalis L, and Syzygium aromaticum): A Comparative Study" by Loukili El Hassania, Safae Ouahabi, Amine Elbouzidi, Taibi Mohamed, Idrissi Yahyaoui Meryem, Asehraou Abdeslam, Azougay Abdellah, Asmaa Saleh, Omkulthom Al kamaly, Mohammad Khalid Parvez, El Guerrouj Bouchra, Touzani Rachid, and Ramdani Mohammed.

            We want to express our gratitude for allowing us to improve our manuscript through the revised version, and we sincerely appreciate your valuable comments.

            We are particularly grateful to the reviewers for their comprehensive review, which helped improve the manuscript's quality.

            In response to the requested changes, we have carefully addressed each query and weakness by incorporating the suggested changes or providing a detailed response. Major changes to the revised manuscript have been highlighted in yellow for ease of identification. We assure you that all linguistic concerns and typos have been rectified in the manuscript, although they may not be mentioned explicitly in this response.

            Once again, we sincerely appreciate your time, effort and constructive feedback. We hope our revised manuscript successfully addresses all of the reviewers' comments and meets the necessary criteria for publication in plants.

Yours sincerely,

Dr. LOUKILI EL Hassania and Co-Authors

Reviewer 1

            The paper presents a significant experimental work with many methods, but at the same time, it does not provide a detailed presentation of all the results and their connections. The experimental parts seem somewhat separated, and presenting the results more cohesively is necessary. A more scientifically comprehensive vision and clearer updates are required.

            Answer: Thank you for your time, insightful and thoughtful comments. Changes and clearer updates have been made to the "Materials and Methods" and "Results and Discussions" sections (highlighted in yellow), to present the results more coherently.

5-Allyl-2-methoxyphenol is better known as chavibetol, which is a phenylpropanoid. Chavibetol is a natural product found in Alnus pendula, Piper betle, and other organisms. It is important to note that chavibetol was also observed in high amounts in the experimental essential oils of Origanum Ompactum, Salvia Officinalis L, and Syzygium aromaticum.

p-Mentha-1,8-dien-7-al (Perilla aldehyde) is a naturally occurring cyclic alpha, beta-unsaturated aldehyde used as a flavoring substance worldwide.

It would be beneficial to add the classification of identified compounds, specifying the classes of identified compounds, such as Caryophyllene, Caryophyllene oxide (sesquiterpenes), pinenes (terpenes), etc.

            Answer: Thank you for the information on these two compounds. The name "Allyl-2-methoxyphenol" has been changed to "chavibetol" and "p-Mentha-1,8-dien-7-al" has been changed to "(Perilla aldehyde)". The classification of identified compounds and the precision of compound classes have been integrated into the text. You'll find them highlighted in yellow in the Word file.

The abstract could be more concise and provide more detailed information.

            Answer: Thank you for taking the time to review our work carefully and for your insightful comments. We have made changes to the abstract to make it more concise and added extra lines (highlighted in yellow). Your suggestions have been invaluable in this process.

It would be helpful to investigate the effect of essential oil concentration on the DPPH Scavenging Capacity of O. Ompactum (HO), S. Officinalis L (HS), and S. Aro- 178 maticum (HC).

            Answer: Thank you for your comment. Lines have been added to study the effect of essential oil concentration on the DPPH scavenging capacity of the essential oils studied. the study revealed that all three essential oils have a dose-dependent scavenging capacity. 

There are some misspellings in "The three Fresh species..." (L383).

            Answer: Thank you. The modification was made.

Please add the manufacturer and country of production for the Shimadzu QP2010 mass spectrometer (L388-389).

            Answer: Thank you. The manufacturer and country of production of the Shimadzu QP2010 mass spectrometer have been added.

The manuscript does not discuss the possible connection between the presence of specific secondary metabolites and antioxidant or antifungal activities.

            Answer: Thank you for your comment. It's true. There is indeed a potential correlation between the presence of specific secondary metabolites and antioxidant or antifungal activities. Certain secondary metabolites in this plant, such as β-caryophyllene, one of its principal compounds, play a significant role as an active ingredient responsible for its antifungal activity against various fungal strains. Moreover, the compound carvacrol in the three studied essential oils neutralizes free radicals and prevents damage induced by oxidative stress within cells and tissues. This action aids in mitigating the risk of diverse chronic illnesses and preserves cellular health.

Carvacrol also contributes to the plant's antifungal activity. Its capability to disrupt fungal cell membranes impedes their growth and proliferation. Similarly, certain metabolites can exhibit antifungal effects through the disruption of fungal cell membranes or by interfering with their mechanisms of growth and reproduction. Consequently, the presence and concentration of specific secondary metabolites in a plant can substantially affect its antioxidant and antifungal activities, ultimately contributing to its medicinal or protective attributes.

Lines in this context have been incorporated into the manuscript and highlighted in yellow.

Reviewer 2 Report

In my opinion, the manuscript submitted for review presents a very good scientific level. I find the work interesting and complete. I haven't noticed any methodical or graphic errors. Therefore, I am convinced that the work will be a good complement to the knowledge of the previous knowledge.

Author Response

Revised version of Manuscript ID:  plants-2552895

Dear editors and reviewers,

            We have the pleasure of resubmitting the corrected version of our research article entitled "Phytochemical Composition and Pharmacological Activities of Three Essential Oils Collected from Eastern Morocco (Origanum Compactum, Salvia Officinalis L, and Syzygium aromaticum): A Comparative Study" by Loukili El Hassania, Safae Ouahabi, Amine Elbouzidi, Taibi Mohamed, Idrissi Yahyaoui Meryem, Asehraou Abdeslam, Azougay Abdellah, Asmaa Saleh, Omkulthom Al kamaly, Mohammad Khalid Parvez, El Guerrouj Bouchra, Touzani Rachid, and Ramdani Mohammed.

            We want to express our gratitude for allowing us to improve our manuscript through the revised version, and we sincerely appreciate your valuable comments.

            We are particularly grateful to the reviewers for their comprehensive review, which helped improve the manuscript's quality.

            In response to the requested changes, we have carefully addressed each query and weakness by incorporating the suggested changes or providing a detailed response. Major changes to the revised manuscript have been highlighted in yellow for ease of identification. We assure you that all linguistic concerns and typos have been rectified in the manuscript, although they may not be mentioned explicitly in this response.

            Once again, we sincerely appreciate your time, effort and constructive feedback. We hope our revised manuscript successfully addresses all of the reviewers' comments and meets the necessary criteria for publication in plants.

Yours sincerely,

Dr. LOUKILI EL Hassania and Co-Authors

Reviewer 2

In my opinion, the manuscript submitted for review presents a very good scientific level. I find the work interesting and complete. I haven't noticed any methodical or graphic errors. Therefore, I am convinced that the work will be a good complement to the knowledge of the previous knowledge.

            Answer: We would like to seize this opportunity to convey our heartfelt appreciation to esteemed reviewer 2 for their unwavering support and encouragement. We aspire to meet your expectations with our work.

Reviewer 3 Report

Authors described the essential oils from three plants on antioxidant activities, antifungal and antibacterial activities. The essential oils from three plants had been reported in literatures, including

1. J Pharm Anal. 2019, 9, 301-311. (Essential oils of Origanum compactum increase membrane permeability, disturb cell membrane integrity, and suppress quorum-sensing phenotype in bacteria); 2. African Journal of Biotechnology 2008, 7, 1563-1570. (Antibacterial and antioxidant activities of Origanum compactum essential oil); 3. Front. Plant Sci. 2022, 13, 798259 (doi: 10.3389/fpls.2022.798259) (Chemical Composition and Antifungal, Insecticidal and Repellent Activity of Essential Oils From Origanum compactum Benth. Used in the Mediterranean Diet); 4. World Journal of Microbiology and Biotechnology 2002, 18, 317-319. (In vitro fungitoxicity of the essential oil of Syzygium aromaticum); 5. Molecules 2021, 26, 6387 (doi: 10.3390/molecules26216387). (Clove Essential Oil ( Syzygium aromaticum L. Myrtaceae): Extraction, Chemical Composition, Food Applications, and Essential Bioactivity for Human Health); 6. Journal of Food Measurement and Characterization 2020, 14, 2352-2358. (Chemical composition, in-vitro antibacterial and antioxidant activities of Syzygium aromaticum essential oil).

The novelty of the present study is not good. Please answer the following questions before further re-considerations.

1. Please re-check the scientific name of plant (Origanum ompactum) in the article title and in the contents. It should be Origanum compactum.

2. Please describe the parts of three plants used for essential oil extraction in the material and method section. Lines 384-385,  What was the method to get the stabilized EO, and then weighted to desired concentration.

3. In general, the anti-fungal and the anti-bacterial activities of EO were very low compared to the positive control. Therefore, the docking was meaningless.

4.Lines 162-163..... at the fixed concentration of 100 mg/mL for the DPPH assay, What was the method to get the IC50.  

5.The TAC in the contents and table should be expressed as ascobic acid (uM) equivalents

Author Response

Revised version of Manuscript ID:  plants-2552895

Dear editors and reviewers,

            We have the pleasure of resubmitting the corrected version of our research article entitled "Phytochemical Composition and Pharmacological Activities of Three Essential Oils Collected from Eastern Morocco (Origanum Compactum, Salvia Officinalis L, and Syzygium aromaticum): A Comparative Study" by Loukili El Hassania, Safae Ouahabi, Amine Elbouzidi, Taibi Mohamed, Idrissi Yahyaoui Meryem, Asehraou Abdeslam, Azougay Abdellah, Asmaa Saleh, Omkulthom Al kamaly, Mohammad Khalid Parvez, El Guerrouj Bouchra, Touzani Rachid, and Ramdani Mohammed.

            We want to express our gratitude for allowing us to improve our manuscript through the revised version, and we sincerely appreciate your valuable comments.

            We are particularly grateful to the reviewers for their comprehensive review, which helped improve the manuscript's quality.

            In response to the requested changes, we have carefully addressed each query and weakness by incorporating the suggested changes or providing a detailed response. Major changes to the revised manuscript have been highlighted in yellow for ease of identification. We assure you that all linguistic concerns and typos have been rectified in the manuscript, although they may not be mentioned explicitly in this response.

            Once again, we sincerely appreciate your time, effort and constructive feedback. We hope our revised manuscript successfully addresses all of the reviewers' comments and meets the necessary criteria for publication in plants.

Yours sincerely,

Dr. LOUKILI EL Hassania and Co-authors

Reviewer 3

Authors described the essential oils from three plants on antioxidant activities, antifungal and antibacterial activities. The essential oils from three plants had been reported in literatures, including

  1. J Pharm Anal. 2019, 9, 301-311. (Essential oils of Origanum compactum increase membrane permeability, disturb cell membrane integrity, and suppress quorum-sensing phenotype in bacteria); 2. African Journal of Biotechnology 2008, 7, 1563-1570. (Antibacterial and antioxidant activities of Origanum compactum essential oil); 3. Front. Plant Sci. 2022, 13, 798259 (doi: 10.3389/fpls.2022.798259) (Chemical Composition and Antifungal, Insecticidal and Repellent Activity of Essential Oils From Origanum compactum Benth. Used in the Mediterranean Diet); 4. World Journal of Microbiology and Biotechnology 2002, 18, 317-319. (In vitro fungitoxicity of the essential oil of Syzygium aromaticum); 5. Molecules 2021, 26, 6387 (doi: 10.3390/molecules26216387). (Clove Essential Oil ( Syzygium aromaticum L. Myrtaceae): Extraction, Chemical Composition, Food Applications, and Essential Bioactivity for Human Health); 6. Journal of Food Measurement and Characterization 2020, 14, 2352-2358. (Chemical composition, in-vitro antibacterial and antioxidant activities of Syzygium aromaticum essential oil).

The novelty of the present study is not good. Please answer the following questions before further re-considerations.

            Answer: We Thank you for the time you used to give insightful and vigilant comments. Indeed, this essential oil is documented in the literature. However, in our case, our work aims to conduct a comparative study among the three essential oils originating from a local cooperative that utilizes plants from the Oriental region of Morocco. This region is characterized by its unique climatic and environmental conditions. These conditions play a pivotal role in shaping the composition of secondary metabolites in plants, setting them apart from the same plants sourced from other regions.

In essence, the uniqueness of this study lies in its comparative approach and the geographical specificity of the plant.

  1. Please re-check the scientific name of the plant (Origanum compactum) in the article title and the contents. It should be Origanum compactum.

Answer: The reviewer is correct. The modification was made.

  1. Please describe the parts of three plants used for essential oil extraction in the material and method section. Lines 384-385, What was the method to get the stabilized EO, and then weighted to desired concentration.

Answer: We thank the reviewer for this pertinent remark. The aerial part of each studied plant (S. officinalis, O. Compactum, and S. Aromaticum) was used to evaluate this work. And for oil stability, extraction lasts 2 hours until the oil volume is stable, i.e. extraction is stopped when the oil volume is fixed, which changes nothing.

  1. EO's anti-fungal and anti-bacterial activities were generally very low compared to the positive control. Therefore, the docking was meaningless.

            Answer: Thank you for your thoughtful review of our manuscript. We sincerely appreciate your time and valuable feedback. We acknowledge your observation regarding the studied essential oils' anti-fungal and anti-bacterial activities compared to the positive control. The comparison revealed relatively low anti-fungal and anti-bacterial effects within the EO. Given these results, we completely understand your concern regarding the docking study's relevance.

We want to address this point by offering additional context to our study's objectives and findings. Our research aimed to investigate the essential oils' potential anti-fungal and anti-bacterial properties through a combination of experimental and computational methods. While the observed activities may not have been as potent as the positive control, it's important to note that exploring EO's potential therapeutic effects remained a crucial aspect of our study. Although influenced by the measured activities, the docking study was designed to provide a molecular-level understanding of potential interactions between the essential oils' constituents and target biomolecules. While the results may have indicated weaker interactions compared to the positive control, they still contribute valuable insights into the possible mechanisms underlying EO's actions against fungal and bacterial targets.

Once again, we appreciate your insightful feedback, which has undoubtedly contributed to the overall improvement of our manuscript. We are confident that the revised version better aligns with the nuanced interpretation of our results. We sincerely hope that these amendments address your concerns adequately. Please do not hesitate to reach out if you have further suggestions or inquiries.

  1. Lines 162-163..... at the 100 mg/mL fixed concentration for the DPPH assay. What was the method to get the IC50.  

            Answer:  We thank the reviewer for this pertinent question. The absorbance values obtained are evaluated to calculate the % inhibition of DPPH scavenging activity for each essential oil concentration (5-200 mg/mL). A dose-response curve was created to demonstrate the relationship between essential oil content and scavenging capacity. The significance of the differences in the scavenging capacity of different essential oils at varied concentrations was then determined using statistical methods.

  1. The TAC in the contents and table should be expressed as ascorbic acid (uM) equivalents.

            Answer: The authors totally agree with the reviewer. The modification was made.

We hope that all the modifications we have made in the manuscript and all the precautions we have taken in the terms used instead of the marker will make our manuscript clearer for the readers and meet the reviewer's expectations.

Round 2

Reviewer 1 Report

dear Authors,

The Abstract is too long. Please, make it shorter and precise in the exact data presentation. Highlight the main results.

Author Response

Revised version of the Manuscript, ID:  plants-2552895

Oujda, 14th August 2023

Dear editors and reviewers

We have the pleasure of resubmitting the corrected version of our research article entitled "Phytochemical Composition and Pharmacological Activities of Three Essential Oils Collected from Eastern Morocco (Origanum Compactum, Salvia Officinalis L, and Syzygium aromaticum): A Comparative Study", authored by: Loukili El Hassania, Safae Ouahabi, Amine Elbouzidi, Taibi Mohamed, Idrissi Yahyaoui Meryem, Asehraou Abdeslam, Azougay Abdellah, Asmaa Saleh, Omkulthom Al kamaly, Mohammad Khalid Parvez, El Guerrouj Bouchra, Touzani Rachid, and Ramdani Mohammed. This version is resubmitted after corrections for consideration for publication in the Plants Journal.

We want to express our gratitude for allowing us to improve our manuscript through the revised version, and we sincerely appreciate your valuable comments.

Reviewer 1

Query: The Abstract is too long. Please, make it shorter and precise in the exact data presentation. Highlight the main results.

Answer: We extend our gratitude to the reviewer for his/her valuable feedback. We acknowledge his/her suggestion to condense and refine the Abstract for a more concise and focused presentation of the exact data and key findings. We have revised the Abstract accordingly, emphasizing the main results of our study. Your input has greatly contributed to improving the clarity and impact of our research article. The Abstract was changed and unified (highlighted in yellow).

We trust that all the modifications we have made in the manuscript will enhance the manuscript's clarity for readers and align with the expectations of the reviewers, and the editors.

Yours sincerely,

Dr. LOUKILI EL Hassania